# An Introduction to Relevant Immunology Principles with Respect to Oral Vaccines in Aquaculture

**DOI:** 10.3390/microorganisms11122917

**Published:** 2023-12-04

**Authors:** Evan M. Jones, Kenneth D. Cain

**Affiliations:** Department of Fisheries and Wildlife, University of Idaho, Moscow, ID 83844, USA; jone3329@vandals.uidaho.edu

**Keywords:** oral vaccine, aquaculture, mucosal immunology, MALT

## Abstract

Vaccines continue to play an enormous role in the progression of aquaculture industries worldwide. Though preventable diseases cause massive economic losses, injection-based vaccine delivery is cost-prohibitive or otherwise impractical for many producers. Most oral vaccines, which are much cheaper to administer, do not provide adequate protection relative to traditional injection or even immersion formulas. Research has focused on determining why there appears to be a lack of protection afforded by oral vaccines. Here, we review the basic immunological principles associated with oral vaccination before discussing the recent progress and current status of oral vaccine research. This knowledge is critical for the development and advancement of efficacious oral vaccines for the aquaculture industry.

## 1. Introduction

The first recorded study of disease prevention using vaccines in fish was likely in 1938, where common carp (*Cyprinus carpio*) were protected against disease after immunization with *Aeromonas caviae* [1]. This was followed closely by another study in 1942 that demonstrated protection in cutthroat trout (*Oncorhynchus clarkii*) against *Aeromonas salmonicida* infection [2]. In the 1980s, Norway’s aquaculture industry experienced a devastating loss due to disease caused by *Vibrio salmonicida* [3]. Since 1988, most salmonids in Norway continue to be vaccinated against this pathogen, and, as a result, mortality and antibiotic use have been greatly reduced, even though production has increased [4]. As aquaculture production has increased, so has the research and availability of fish vaccines. New biotechnological tools and research have allowed for the development of a variety of vaccine types including killed, live-attenuated, subunit, DNA, and RNA-based vaccines [5]. Injection vaccines are the current industry standard because they generally offer the most robust protection against disease due to the direct stimulation of the systemic immune system. This protection comes at the expense of vaccine administration being more stressful for fish and labor-intensive for the workforce. Immersion vaccines were developed to resolve some obstacles associated with injection vaccines but tend to offer a reduced efficacy and an increased difficulty in incorporating adjuvants. Unlike injection vaccines, immersion vaccines can be administered to fish smaller than approximately 20 g; this is important because most mortality is experienced in young fish whose immune systems have not adapted to pathogens in the rearing environment. Oral vaccines take this progress a step further, essentially eliminating the requirement for extra labor because vaccine administration is similar to regular feeding. Several important issues hinder the widespread use of oral vaccines in the industry, and the root causes and progress associated with these issues will be discussed throughout this review. Though the commercialization of new types of vaccines is an arduous process [6], they remain one of the most important tools in the aquaculture industry and are an essential component to any large-scale intensive culture facility.

For oral vaccines, the major immunological component of interest is the gut-associated lymphoid tissue (GALT), because it is the first immune system structure exposed to the vaccine and, thus, key for antigen absorption. Humoral, or antibody-mediated, immunity remains the most common parameter measured to evaluate vaccination success. Cell-mediated immunity, conversely, is measured less often due to a relative lack of tools for identifying surface markers related to T cells in many fish species, the foundation of this arm of the adaptive immune response. Monoclonal antibodies (mAbs) specific to fish immunoglobulin exist for a variety of species and are a necessary tool for measuring the adaptive immune response. To provide a thorough description of oral vaccination fundamentals, we began our review effort searching for papers primarily focused on recent characterizations of fish humoral and mucosal immunity. We then searched for papers describing specific aspects of oral vaccination and included key words in our search such as intestinal immunology, encapsulation, antigen preparation, delivery platform, immune response, and disease protection. The structure of this paper follows our research effort, first reviewing general fish immunity before discussing research specific to oral vaccination.

## 2. Principles of Immunology Relevant to Oral Vaccination

### 2.1. Innate Immunity

Innate immunity refers to the nonspecific, immediate immune response against infections in an animal. Activation of the innate immune response is not the main focus when developing and testing vaccines, but this arm of the immune system still plays a vital role in a vaccine’s success. The first task of the innate immune system is sensing a potential infectious agent by identifying pathogen-associated molecular patterns (PAMPs) through innate cells’ pathogen recognition receptors (PRRs) [7]. Once pathogens are identified, the innate immune system stimulates the production of chemokines and cytokines responsible for the inflammatory response and upregulation of the stimulatory molecules for T cell responses [8,9]. In the context of oral vaccines and other mucosal vaccines, the innate immune system presents a challenge of tolerance. The mucosal barriers are constantly presented with antigens and stimuli from commensal or foreign microbes; the mingling of ‘self’ and ‘non-self’ is so commonplace that inducing an immune response for each instance would lead to severe inflammation of the mucosal barriers and negative impacts on the animal’s health. Therefore, the innate immune system must operate with some tolerance towards these antigens. Meanwhile, mucosal vaccines, including oral vaccines, must find a way to break through this tolerance to provoke a robust immune response that will provide protection against a specific pathogen [10]. The tolerance of gut mucosal immune systems will be discussed later in this paper. 

### 2.2. Adaptive Immunity

Immunology is an important field of vaccinology, with a special focus on the activation of the adaptive immune response. However, it is important to note that, without the innate immune response’s ability to recognize and signal potential infectious bodies within a host, the adaptive response would be ineffective. The adaptive response is characterized by a slow response that can take days or weeks to manifest and is highly dependent on water temperature, with lower temperatures being associated with a slower response [11,12]. The adaptive immune response can be broadly split into the following two categories: humoral and cell-mediated responses [13]. The cell lineage vital to this process is the lymphocyte, divided into B (humoral) and T (cell-mediated) cells [14]. In teleost fish, both T and B cells originate in the head kidney; B cells also mature here, while T cells migrate to the thymus for maturation [15,16]. 

B cells are responsible for the humoral, or antibody-mediated, immune response. To express and secrete antibodies, B cells are first exposed to antigens, with the help of the innate immune system and antigen-presenting cells (APCs). B cells then develop into plasmablasts and eventually mature plasma cells secreting specific antibodies [17]. B cells and their antibodies are important defenders and can combat pathogens through direct neutralization as well as through indirect mechanisms such as opsonization, complement activation, and antigen presentation to other immune cells [18].

Three distinct types of immunoglobulins have been described in teleost fish, each following the basic structure of two heavy chains and two light chains: IgM, IgD, and IgT/IgZ [19]. The most abundant immunoglobulin in fish, IgM, was the first type described in fish and can be found in almost all vertebrates [20]. IgM can be found on B cell membranes (membrane-bound) and in its secreted tetrameric form (secretory) in blood circulation and mucus. Some amount of IgM can be detected in circulation without antigen stimulation, but its levels are increased after antigen exposure and subsequent immune system activation [19]. Studies have also shown that the secreted tetrameric form of IgM can be transported into mucosal tissues, including skin mucus and other mucosal tissues discussed later in this review [21]. This antibody class assists with a wide range of pathogen defenses, including complement activation, agglutination, and neutralization [18,19,22]. As the most abundant immunoglobulin, it is a popular parameter measured in fish vaccination studies to evaluate immune stimulation and predict efficacy [23,24,25]. 

Other immunoglobulin classes are not nearly as well-described in function or origin relative to IgM. IgD was first discovered in channel catfish (*Ictalurus punctatus*) in 1997 [26], and its functions remain somewhat of a mystery, even in mammalian species [19]. There is evidence of B cells expressing both IgM and IgD, as well as solely IgD [27]. IgD is found in several other fish species [28,29,30], and the highest levels are generally found within the head kidney tissue [31]. A monoclonal antibody specific to rainbow trout (*O. mykiss*), IgD was used to measure levels within sera, when researchers found that its levels were up to 400 times lower relative to the IgM in the sera [32]. IgT (known as IgZ in zebrafish (*Danio rerio*)) was the focus of several recent studies aimed at characterizing the mucosal immunity of fish [27]. It was first discovered in rainbow trout and zebrafish in 2005, and no ortholog has been found in mammals or birds to date [33,34]. One study found that specific IgT antibodies were increased in the mucosal barriers of the gastrointestinal tract, but sera antibodies were dominated by IgM during a parasitic infection [27]. Bath and oral vaccinations that act on the mucosal barriers of the skin and intestinal tract have been shown to induce greater levels of IgT expression in tissues for both bacterial and viral antigens [35,36]. To date, no B cells have been identified that jointly express IgT and IgM, indicating a separate development and lineage for each isotype [18]. The hypothesis of separate lineages is further supported by the lack of evidence demonstrating class-switching capabilities of Igs in teleost fish [37]. The results from these studies are only possible because monoclonal antibodies (mAbs) have been developed that are specific to various Ig forms in teleosts.

Cell-mediated immunity is another important component of the immune response, particularly when combatting intracellular pathogens which have escaped antibody-mediated defense mechanisms. The cell-mediated immune response is governed primarily by T lymphocytes. There are two broad types of T cells: γδ and αβ T cells. The γδ T cells behave similarly to innate pathogen recognition cells in that they do not require antigen processing and presentation through the normal signaling proteins. The function of γδ T cells in fish is not fully understood, but it is hypothesized that they are involved in antigen recognition and may play an important role in mucosal immunity [38]. This is supported by an increased prevalence of γδ T cells in mucosal and epithelial tissues, the barriers at which pathogens are first met with resistance from the host [7]. However, studies identifying and characterizing the function of this type of T cell are lacking. To date, the characterization of γδ T cells has largely used genomic and molecular techniques to identify genes and characterize their expression [39]; antibody tools have only been used for zebrafish [40,41].

The other form of T cells, αβ T cells, are better characterized in fish immunology and vaccinology, particularly for viral vaccines. The αβ T cells are divided into cytotoxic T lymphocytes (CTLs) or T helper cells (T_H_). CTLs are directly responsible for killing infected or abnormal cells within the host, based on the antigens presented on the surface of the targeted cells. Viral pathogens typically provoke antigen presentation on the outside of host cells with major histocompatibility complex (MHC) molecules, which CTLs target. Thus, for many anti-viral vaccines, CTL activity is measured to characterize the immune response. Several anti-viral oral vaccines have characterized CTL responses, including inactivated preparations [42], live-attenuated [43], and DNA vaccines [44,45]. A recent review of teleost CTL responses to infection and vaccination by Yamaguchi et al. provides more detailed information for studies on viral infection and vaccination [46]. CTLs are often identified through the CD8 marker on their outer membrane, which assists with cell-to-cell interactions and signaling with MHC-1 proteins; T_H_ cells are identified by their presence of CD4 on their outer membrane, which mediates interactions with MHC-II proteins [7]. Genetic expression of these proteins, CD8 and CD4, is the most common way of identifying and characterizing these populations of T cells; antibody tools targeting the cellular characterization of T cells have yet to be developed for many teleost species [16]. T_H_ cells assist the cell-mediated and humoral immune response by producing cytokines that act as signalers to other immune cells [47,48]. These cytokines can play important roles in stimulating the humoral immune system and provoking stronger antibody responses to viral antigens, demonstrating the cooperation between the cell-mediated and humoral arms of the immune system [49]. The use of antibodies specific to teleost CD4 have helped characterize T_H_ cell responses to viral antigens in rainbow trout, zebrafish, and olive flounder (*Paralichthys olivaceus*), according to a recent review of T_H_ cell responses [50], but continued development of these tools is necessary for a deeper understanding of T_H_ cell function and response to various stimuli. 

### 2.3. Mucosal Immunology

Fundamentally, the mechanisms of vaccines stimulating a systemic immune response are similar across injection and mucosal delivery methods. APCs present the vaccine antigen through their MHC-II class molecules, a step which activates T cells and stimulates B cell proliferation and antibody secretion [51]. The major differences in the response to the vaccine between the two delivery methods are the following: where the antigens are introduced, and, subsequently, where the immune response radiates from. For injection vaccination, the antigen bypasses all mucosal defenses of the host fish; this results in a robust immune response that causes IgM antibodies to circulate throughout the serum of the fish [52]. Unlike mucosal vaccines, there is no commensal microbe population to account for in an injected vaccine, so antigens are directly exposed to the host cells. Oral and immersion vaccines aim to stimulate an immune response in specific tissues that are most likely to encounter target pathogens, with the added benefit of stimulating IgT in addition to IgM. These tissues are characterized by a complex equilibrium of host and commensal microbe communities that is still not fully understood [53]. Immersion vaccines can stimulate the production of mucosal antibodies in skin and gill tissue, which ideally protects the fish from pathogens crossing these barriers and causing disease. The stimulation of mucosal antibodies was demonstrated for several antigen targets, though protection against pathogen exposure was variable [52]. Oral vaccines stimulate immune responses local to the intestinal tract, but have a more complex antigen uptake route compared to the injection or immersion methods because of the intestinal transit of the vaccine [52]. The complexities and associated challenges of oral vaccine antigen uptake will be discussed in later sections, after other mucosal tissue immunity is reviewed. 

#### 2.3.1. Organized Mucosal Lymphoid Tissues

Until recently, it was thought that most fish lack traditional, organized structures of lymphoid tissues found in other animals, such as lymph nodes and Peyer’s patches [54]. These organized structures are identified and characterized by the aggregation of immune cells, including B and T cells, and a marked change in structure during an immune response. In 2008, researchers found evidence of organized lymphoid tissue in the gills of Atlantic salmon (*Salmo salar*), naming it interbranchial lymphoid tissue or ILT [55]. Evidence of similar ILT-like structures was also found in several eel and carp species, including zebrafish; in each of the species tested, the structure and cellular makeup was similar to other mammalian secondary lymphoid organs [56]. Interestingly, organized ILT structures were not found in several spiny-rayed fish species tested. The authors hypothesized that ILT closely co-evolved with interbranchial septa due to the location of the ILT at the base of interbranchial clefts. The very short interbranchial septa in spiny-rayed fishes such as the perch and the flounder, coupled with the lack of a traditional ILT structure in these fish, supports this hypothesis [56]. Another organized structure was found along the sides of each gill arch at the base of the filament in zebrafish; this amphibranchial lymphoid tissue (ALT) shares similarities in its structure and response to infection [57]. Unlike the ILT, this ALT structure was present in spiny-rayed fishes, while other species with longer interbranchial septa had both ILT and ALT, including cyprinids [57]. Yet another lymphoid structure found in Atlantic salmon in an area analogous to the cloaca of birds has recently been characterized. This bursa is described as potentially performing secondary lymphoid functions and has a higher proportion of T cells relative to B cells [58]. To date, the origin of this structure is not fully understood, and its presence in other teleost species remains unknown. Another example of organized lymphoid structure is an aggregation of lymphocytes in the nasal cavity, which has only been described in rainbow trout thus far and has been designated as organized nasal-associated lymphoid tissue (O-NALT) [59]. The same study found a proliferation of B cells in the O-NALT after nasal vaccination, while other areas of the nasal cavity without known organized lymphoid structures did not demonstrate the same type of response [59]. Finally, this year, a pre-print manuscript was made available, describing an organized lymphoid structure named the Nemausen Lymphoid Organ (NEMO), found in the branchial cavity of teleost fish; this may be part of a larger immune complex including the ILT and ALT [60]. The recent increase in the identification of organized lymphoid structures over the last five years demonstrates how little is known about some areas of the teleost immune system. It is possible, if not likely, that organized lymphoid structures exist within the intestinal tract and may play an important role in the immune response to oral vaccination, though further studies are required to provide a clearer picture.

#### 2.3.2. Diffuse Mucosal Lymphoid Tissues

Diffuse networks of leukocytes in mucosal barriers are more commonly described in the teleost vaccine literature; these are generally called mucosa-associated lymphoid tissues (MALTs). These tissues have been described in several mucosal sites of fish, including skin (SALT), gill (GiALT), gut (GALT), nasal (NALT), buccal, and pharyngeal sites, but are likely present in all fish mucosal barriers [61,62]; the main MALTs are shown in Figure 1, along with the organized lymphoid structures described previously. Characteristic of mucosal barriers, these lymphoid tissues are colonized with a diverse arrangement of commensal microbes, and immune responses must be carefully regulated to provide defense while avoiding over-stimulation [53].

Each MALT responds differently to different routes of vaccination. Mucosal vaccination routes and the MALTs stimulated by each are shown in Table 1. The immune function of the NALT was the subject of several recent papers on rainbow trout. Evidence shows that a local and systemic response can be stimulated through nasal vaccination [63,64] and in response to parasitic infection [65]. It is hypothesized that immersion vaccination also stimulates a local nasal response in fish, but further research is needed to determine the extent of local stimulation [66]. The epidermis of fish contains mucus-secreting cells, making it the largest diffuse lymphoid tissue. IgM plays a major role in the SALT, but the IgT levels are much higher in the skin mucus relative to the sera [67,68]. The buccal and pharyngeal ALTs are relatively new discoveries in rainbow trout, with both being characterized by a strong IgT response after parasitic infection, with only limited IgM and IgD detection [69]. The GiALT is unique to fish and also contains organized lymphoid structures, the ILT and the ALT, mentioned in the previous section. These structures are characterized by aggregations of T cells and are thought to be important in antigen encounters, but not a site of maturation or development of immune cells. Though research on diffuse lymphoid tissues is more established relative to that on organized teleost lymphoid tissues, it is still new information when compared to research on localized immune structures such as the thymus or the spleen. There remain knowledge gaps in how these diffuse tissues balance the mucosal environment with commensals and pathogens. Microbiome research is outside the scope of this review but may become an important field of study for vaccinology, particularly oral vaccines, as the role of the microbiome in teleost immunology becomes more defined. 

### 2.4. Intestinal Immunology

In terms of oral vaccination, the gut-associated lymphoid tissue (GALT) is the most important site of immune activation. This may also be the most well-researched ALT in fish and was first reviewed in 1988 [73]. Gut B cells, both IgM+ and IgT+, are primarily found in the lamina propria, with IgT+ cells making up 54% of the total resident B cells in rainbow trout [27]. The presence of immunoglobulins in the gut mucosa is difficult to measure in part due to the difficulty of obtaining high quality samples [74]. Research has demonstrated that sera IgM quickly degrades in the presence of gut mucus, but not skin mucus, likely because of the high proteolytic activity [75]. Several authors have also hypothesized that there are subtle differences in the structure or specificity of sera and mucus IgM and that current mAbs used to identify Igs may not be sensitive to such small differences [27,74,76]. For instance, one study found that an mAb could be specific to the heavy (H) chain of mucus IgM, but not sera IgM, in carp [75]. There is still difficulty when using tools to measure specific IgT levels in mucosal samples. Much of the research has focused on rainbow trout, in which the isotype was originally discovered, and on mucosal surfaces other than the gut [65,77,78]. Recently, an mAb was developed for IgT in yellow croaker fish (*Larimichthys crocea*), which will likely provide a useful comparison as IgT and other mucosal immune responses are found in multiple fish species [79]. The design of efficacious vaccines for any delivery route in aquaculture depends on the advancement of immunological tools, such as mAbs, that can better characterize the immune response of emerging aquaculture species. These tools can define the cellular interactions within the intestinal tract more clearly, as well as the subsequent effect on the systemic immune response. 

A major role of the GALT is its ability to absorb antigens and present them to cells that subsequently stimulate the immune response; this was first reported in common carp [76,80] and, later, in sea bass (*Dicentrarchus labrax*) [81]. In most fish, the gastrointestinal tract can be divided into three segments (Figure 1), as follows: the first is dedicated to protein absorption [82]; the second specializes in macromolecule uptake [83]; and the full function of the third segment is still debated but likely involves osmoregulation [62] and, ostensibly, is not involved in nutrient absorption due to limited presence of microvilli [84]. Antigen absorption is required to stimulate an effective immune response after oral vaccination, as, without the presentation of antigens, the immune system lacks recognizable targets during later infections, and the benefits of vaccination are not fully realized. This also requires antigen to remain intact, maintaining PAMPs for the immune system to recognize. This is a major obstacle for oral vaccination because of the harsh environment of the gastrointestinal tract, which is designed to break materials down to prepare them for absorption. To solve this issue, various forms of encapsulation have been used to deliver intact antigens to the areas of the gut responsible for absorption. 

A variety of encapsulation technologies have been used for oral vaccines in fish; some of the more common approaches include the use of alginate, chitosan, liposomes, or Poly D,L-lactic-co-glycolic acid (PLGA) for antigen protection [85,86]. Alginate is a natural polysaccharide from the cell walls of various brown algae species that is a relatively inexpensive ingredient; most importantly, it is stable at low pH levels, such as those found in the stomach, and readily disintegrates at the neutral–basic pH levels encountered in the hindgut of fish [87]. Chitosan is more often used in the preparation of nanoparticles carrying oral vaccines and is derived from the chitin of crustaceans and insects. Chitosan has adjuvant properties that stimulate the innate immune response, in addition to the benefits of an increased antigen retention time and the opening of tight junctions between epithelial cells for a more efficient absorption [88]. Liposome encapsulation consists of carrying vaccines in spheres made of phospholipid bilayers capable of carrying hydrophilic and lipophilic formulations [89]. PLGA is a copolymer formed through the binding of lactic and glycolic acids, with different forms of PLGA available depending on the ratio of the acids used during production. PLGA is stable in biological fluids, has some beneficial adjuvant effects, and its long history of testing in mammalian systems has led to its approval from the U.S. Federal Drug Administration (FDA) as a drug delivery compound [85]. Many of these encapsulation techniques can be used in tandem with each other, such as chitosan–alginate particles [90,91,92], and in the formation of nanoparticles [93,94,95,96]. Nanoparticles are valued because of their interaction with immune cell receptors in the host, making them more readily absorbed through endocytosis, their adjuvant effects, and their flexibility in creating particles with different surface properties, shapes, and ingredients [96]. In addition to previously mentioned encapsulation platforms, nanoparticles also lend themselves to the formation of virus-like particles (VLPs), a relatively new oral vaccine platform which has demonstrated the ability to stimulate neutralizing antibodies and may be safer than live-attenuated vaccines in some cases [97]. Encapsulation technology is ongoing, and new combinations of compounds and preparations are being studied each year. It is important to understand that the success of various encapsulation methods is not necessarily measured by the stability of the antigen throughout the entire intestinal tract. Antigens must eventually be released for the host immune system to sample and respond to their presence; therefore, any encapsulation platform must be designed to eventually break down and release its contents for absorption in the intestinal tract. It is likely that a single encapsulation method will not be used exclusively across the commercial aquaculture industry, but that multiple viable methods will be considered useful. Some encapsulation methods may be less expensive to produce but provide slightly worse protection against a disease relative to others; this would be useful for farms that do not experience significant disease losses and grow fish to market-size themselves. Additionally, a cheaper and less protective formulation may be more acceptable as a booster dose, as is the case with some current vaccine programs which use more effective injection vaccines as a primary dose and less effective oral vaccines as a booster dose [98]. Other production systems that generate most of their income from selling juvenile fish to other growers may find it prudent to invest in more expensive encapsulated oral vaccines that provide a better protection; this would help maintain customer satisfaction, particularly when customers may experience a variety of mortality rates at their facilities. Ultimately, studies should continue to evaluate a wide variety of encapsulation methods to provide the industry with multiple options to suit individual needs. Price, ease of manufacturing, and shelf stability are important variables to be considered; research should avoid becoming fixated on pursuing an ideal industry standard that may not exist.

The method and route of antigen absorption depends on the characteristics of the target antigen. Small antigens may be transported directly through the tight junctions of epithelial cells in the gut, known as paracellular transport [70,99]. Transcellular antigen transport refers to the transport of antigens through a cell; examples of this type of transport are widely available in the literature. Transcellular transport may occur in a fluid- or solid-phase uptake. During fluid-phase process, soluble antigens are taken into an intestinal cell with extracellular fluid by pinocytosis [70], while solid-phase uptake can occur either by receptor-mediated endocytosis, if the particulate antigen is small enough, or by phagocytosis, in the case of large antigens [70]. The absorption of both soluble and particulate antigens has been characterized in common carp; researchers found that both antigen types were absorbed in the second gut segment by epithelial cells, then transported through vacuoles before being presented to interepithelial macrophages [100]. They also found specific antibodies in the skin mucus and sera for the particulate antigen, but the soluble antigen only provoked an antibody response in the sera [101]. This demonstrates a common mucosal system throughout the fish, since gut antigen exposure led to skin antibody production, and that the transportation of sera antibodies to mucosal sites is somewhat limited. Differences were also found in the processing time for the antigen types, with the receptor-mediated uptake of particulate antigens transporting them to the blood in as little as 30 min in trout [102] and common carp [103], while the soluble antigen presence in the blood was only observed 4 h after exposure in common carp [101]. These studies also confirmed that the second gut segment was the primary site of immune activation, with large numbers of resident macrophages present in a stable state and smaller, mobile macrophages arriving after antigen exposure [101]. Though, it should be noted that, due to the diversity of teleosts, there are likely differences in the sites and structures important for antigen sampling among species [99]. Several papers include sections dedicated to describing antigen absorption in more detail [70,86,99]. In the future, as vaccine encapsulation technology advances, a better characterization of antigen absorption will aid in the development of efficacious oral vaccines [85].

Antigen tolerance in mucosal barriers is an important aspect of any mucosal vaccine but can pose issues particularly for oral vaccination. As mentioned previously, mucosal barriers are in constant contact with foreign substances, and an immune reaction to each of these encounters would overwhelm the host animal. As such, mucosal barriers have evolved to have a certain amount of tolerance to foreign antigens, particularly those which are encountered repeatedly in low concentrations and do not result in significant cell harm; this has been demonstrated in immersion, oral, and anal delivery of vaccines [101,104,105]. In oral vaccination, this phenomenon is referred to as oral tolerance, and it is thought to be linked to the abundance and variety of commensal microbes colonizing the intestinal tract, as well as the frequent and repeated delivery of target antigens [70,106,107]. The exact mechanisms of this tolerance are not fully understood in fish or mammals, though it is thought to involve regulatory T cells and tolerogenic dendritic cells, a type of antigen-presenting cell [108]. Continued research on the gut microbiome in all animals will help illuminate the mechanisms of tolerance; this is important for the development of efficacious oral vaccine platforms in aquaculture [53]. Early studies on oral tolerance focused on differences in antibody titers after exposure to antigens, with weaker antibody responses observed after multiple antigen exposures in common carp [100,106]. In coho salmon (*O. kisutch*), high and medium concentrations of vaccine led to a weaker specific antibody response relative to the low-concentration treatment group [109]. In rainbow trout, antibody responses were dose-dependent when fish were exposed to the human γ-globulin antigen, but a similar response was not detected after a similar exposure to *A. salmonicida* antigens, demonstrating that the type of antigen, in addition to the duration and dose of exposure, affects tolerance [107]. More recently, a better understanding of immune-related genes has allowed researchers to characterize some of the cell-signaling pathways altered by oral tolerance. The *foxp3* transcription factor is a commonly used marker for immune tolerance as it is associated with regulatory T cells and suppresses cytokine production in leukocytes [110,111]. This gene is downregulated during an active immune response, decreasing regulatory T cell activity and allowing for increased cytokine production. Rainbow trout exposed to low antigen doses in oral vaccines have demonstrated upregulated *foxp3* relative to the treatment groups with higher antigen doses [105]. Finally, as more feed additives and adjuvants are tested for use in aquaculture, it is important to understand how their incorporation into fish diets may affect oral tolerance. Some feed additives result in unintended inflammatory immune responses; to avoid this, fish are selected, in part, based on their tolerance to the feed additive [112]. Future studies of oral vaccination and immune tolerance should account for potential strain differences. The continued growth of the aquaculture industry requires for the development of oral vaccines to consider the progress made in sustainable fish feeds. Changes in fish feed ingredients will have cascading effects on the gut microbiome, function of the GALT, and, ultimately, the efficacy of various oral vaccine platforms. 

## 3. History and Present Status of Oral Vaccination

Oral vaccination refers to any formulated vaccine that is delivered through the buccal cavity into the gastrointestinal tract. The earliest oral vaccination experiment in the literature is by Duff, in 1942, who used a killed *A. salmonicida* preparation to coat feed particles which successfully protected cutthroat trout (*Oncorhynchus clarkii*) in a pathogen challenge [2]. Since then, oral vaccines for fish have advanced in both their preparation and administration, and there are currently several commercial products used to protect against the following pathogens: infectious pancreatic necrosis virus (IPNv), spring viremia of carp virus (SVCv), infectious salmon anemia virus (ISAv), *Piscirickettsia salmonis*, *Y. ruckeri*, *Vibrio* spp., and *Lactococcus garviae* [5,108,113,114]. One of the main obstacles to developing efficacious oral vaccines is protecting the antigen during its passage through the acidic environment of the stomach, until it reaches the second gut segment [86]. This destruction means that oral vaccines often require higher doses of antigen, compared to anal vaccination, to stimulate similar levels of specific IgM. In one case, oral vaccination required a dose 50× higher than anal vaccination to achieve similar specific IgM levels in sera [115]. Though oral vaccination is proven to stimulate an immune response and provide protection against disease, this inefficiency and potential oral tolerance for some antigens make it a less attractive option relative to other vaccine delivery platforms such as injection and immersion.

Some popular materials for vaccine antigen encapsulation were mentioned previously, but how best to use these materials for antigen protection remains an active area of study [93]. An ideal method of delivery is incorporating the vaccine directly into fish feed and capitalizing on the relative ease of vaccine administration by combining it with the routine feeding activities. Vaccines can be included into fish feed by coating the surface of the feed particles directly using oil; this typically involves unencapsulated vaccines in a powdered state or encapsulated formulations [116]. Alternatively, a liquid form of the vaccine can be added to the feed ingredients directly to achieve a homogenous mixture of vaccine throughout the feed particle [117]. Inclusion of the vaccine during the feed formulation process is only possible at its later stages because the temperatures and pressures used early in the process would destroy the vaccine [86]. 

The majority of recent studies on oral vaccination test encapsulation methods in the context of immune stimulation and protection from disease. These studies involve a variety of species and target antigens; even when similar encapsulation materials are used, differences in encapsulation methods or vaccine formulation can generate different results. Microalgae is a relatively new encapsulation technique that is attractive for its wide availability and low costs. One study transformed algae to promote expression of an antigenic protein from *Renibacterium salmoninarum*, the causative agent of bacterial kidney disease, and found that oral vaccination with the algae promoted a specific antibody response [118]. Another study encapsulated green fluorescent protein (GFP) in microalgae and detected intact proteins in the intestinal tissue of zebrafish after oral feeding [119]. Though these results are promising, no evidence of disease protection from a microalgal-encapsulated oral vaccine has been published. Encapsulation within microparticles is one of the most popular strategies for oral vaccine delivery. Alginate is a common material used for microparticle encapsulation, and alginate microparticles (AM) are a promising drug delivery system in animal and human medicine [120]. AMs have been tested for oral vaccination in fish since at least 1994 [90] and include many vaccines against a variety of pathogens such as *V. anguillarum* [121], *A. hydrophila* [122], *L. garvieae* [123], infectious hematopoietic necrosis virus (IHNv) [44], *S. iniae* [92], spring viremia of carp virus (SVCv) [91], infectious pancreatic necrosis virus (IPNv) [124], lymphocystis disease virus (LCDv) [125], and *A. salmonicida* [126]. All these studies were successful in promoting a specific immune response, though the type of vaccine used varied, with DNA vaccines being well-represented [44,91,92,124,125,126] relative to whole-cell or antigen preparations [121,122,123]. Nanoparticles are another promising method of encapsulation and can be formed with many of the encapsulation materials mentioned previously; nanoparticles have also been the subject of several review papers [93,95,96,127]. Nanoparticles containing an inactivated virus reduced mortality in Atlantic salmon challenged with the infectious salmon anaemia virus (ISAv), though an adjuvant within the nanoparticles was required for significant protection [128]. In rohu (*Labeo rohita*), an outer membrane protein vaccine to *A. hydrophila* stimulated specific antibody production and reduced mortality during challenge after the oral administration of nanoparticles [129]. Several studies have obtained a relative percent survival (RPS) greater than 70% using nanoparticles or virus-like particles (VLP) orally administered in a variety of species, including black seabream (*Acanthopagrus schlegelii*) [130], Atlantic salmon [128], rohu [131], and convict grouper (*Hyporthodus septemfasciatus*) [132]. Some microorganisms are hardy enough to weather the digestive environment, and bacteria, yeast, and plant cells are all potential carriers for oral vaccines [108]. Transgenic plants can be designed to express viral proteins that can then be harvested and delivered to fish orally without further encapsulation. One study used transgenic plants to produce the capsid protein of the nervous necrosis virus (NNv), which self-assembled into VLPs; when administered orally, this platform stimulated an immune response in a sevenband grouper (*H. septemfasciatus*) comparable to a commercial, injectable vaccine [133]. Studies of yeast platforms show a potential use for vaccinating larval marine fish; yeast made to express a green fluorescent protein (GFP) was fed to plankton, which were subsequently fed to flounder (*Platichthys flesus*), in which the GFP was still shown intact within the larvae’s intestine; this system was also shown to stimulate a local innate immune response in juvenile rainbow trout, demonstrating a potential adjuvant effect [134]. Other studies have also shown yeast to be an effective oral delivery platform that can protect vaccine antigens and stimulate an adaptive immune response [135,136,137]. Bacteria can also be an effective platform, with many studies demonstrating the potential of *Escherichia coli* as a carrier. A recombinant vaccine against NNv was overproduced in *E. coli* cells that had been incorporated directly into feed and administered to European sea bass, which resulted in an RPS of 100% [138]. Yet another recent study fed *E. coli* cells expressing an SVCv to rotifers, which were then fed to common carp and provided protection relative to the unvaccinated control group [139]. Another oral vaccine bacterial platform that is gaining traction is the use of biofilms to protect a vaccine during gut transit [140]. A recent study demonstrated protection against *Photobacterium damselae* subsp. *damselae* in giant grouper (*Epinephelus lanceolatus*); this platform used a formalin-inactivated biofilm that was incorporated directly into the feed, which provided an RPS of 62% relative to the whole-cell formalin-inactivated vaccine [141]. This same biofilm vaccination method has also demonstrated protection against *L. garvieae* in mullet (*Mugil cephalus*) [142]. These studies are just a few recent examples that demonstrate the variety of encapsulation methods and material combinations that can be used to develop efficacious oral vaccines. Other papers more closely review the pros and cons of various platforms [52,93,96,120,127,143,144]; however, comparing the results of individual studies can be difficult due to differences in the length of vaccine delivery, dosing, vaccine preparation (inactivated, attenuated, recombinant, etc.), and fish size, age, or strain. The lack of a widespread use of oral vaccines may be due, in part, to the huge variety of platforms that have been tested and the differences in immune stimulation and protection; there are so many potential candidates that it can be difficult to select one in which to invest the time and resources needed to develop a commercially viable oral vaccine platform.

Oral vaccines that protect against enteric pathogens, or those that normally colonize and infect the gastrointestinal tract, have had the most widespread success in aquaculture. The first commercial oral vaccine, AquaVac ERM Oral [98], was developed after a long history of successful experimental oral vaccines for rainbow trout, starting in 1965 [145]. This is an inactivated whole-cell bacterium that ultimately offers better protection through injection and immersion routes. Therefore, this commercial product is offered strictly as a secondary booster to the more effective primary routes; this strategy may be the most applicable use for oral vaccines in aquaculture. More recently, there has been progress in formulating a live-attenuated vaccine against *Edwardsiella ictaluri* for use in channel and hybrid catfish (*Ictaluris punctata*) [117,146]. This is a unique and simple preparation, with the live bacterial vaccine diluted in raw well-water before being sprayed onto feed immediately before feeding. Significant protection has been demonstrated with RPS, ranging between 80 and 100% depending on the dose used. Other research into live-attenuated oral vaccines is limited, largely because few live-attenuated vaccines are available for any given immunization route. Instead of coating feed directly, other studies have utilized oral gavage or microparticle encapsulation to administer the vaccines. Oral gavage is not a commercially viable option, but it allows for more control and reproducibility in experiments. In Nile tilapia (*Oreochromis niloticus*), an avirulent *Streptococcus agalacitae* was delivered through oral gavage and provided an RPS of 71% during challenge 15 days after vaccination, but this declined to 53% 30 days post vaccination, and both were outperformed by the injection vaccination [147]. Another study tested alginate microparticles both unencapsulated and encapsulated with a live-attenuated *Flavobacterium psychrophilum* vaccine [148]. The researchers demonstrated that the attenuated vaccine was still viable after encapsulation, though a stronger serum antibody response was observed in the nonencapsulated treatment; the antibody response of the fish vaccinated with the nonencapsulated treatment was equal to that observed in the fish vaccinated by injection. There was evidence of protection during pathogen challenge; however, a high challenge dose overwhelmed protection even in the injection-vaccinated groups, muddling the interpretation of the results. An older study encapsulated a live-attenuated VHSv in lipid particles fed directly to fish, which provided RPS values of 37% and 100% during challenge 28 days after vaccination [149]. However, this study did not report specific antibody responses or the duration of protection. 

## 4. Conclusions

Though the effectiveness of oral vaccines is variable and generally lower than injection or immersion, orally administered vaccines can stimulate immune responses and, in some cases, provide acceptable levels of protection to justify their commercial use. In certain situations, like the live-attenuated *E. ictaluri* vaccine [146], oral vaccines are preferred over immersion, because fish are stocked into ponds prior to immunocompetence. Harvesting fish for vaccination is prohibitively expensive and stressful, reducing the economic returns of vaccination, making oral delivery ideal for pond-raised fish. Similarly, the implementation of oral vaccination is an easier, more efficient vaccination strategy for fish stocked in net pens, since access to the animals is limited. Enteric pathogens, such as *Y. ruckeri* or *E. ictaluri*, are good candidates for oral vaccination because of the reduced complexity of the encapsulation requirements [145]. Vaccine development using novel techniques is expensive, particularly when attempting to change longstanding industry standards; it requires the new technology to drastically outperform the existing techniques or for it to be significantly less expensive, without a severe decrease in efficacy. Though few oral vaccines have widespread use thus far, the potential benefits provide incentive enough for continued research and development. This review provides fundamental information required to begin research into oral vaccination, though it is not exhaustive. Other topics such as the interactions of the intestinal microbiome and biochemistry of the vaccine materials with host and commensal cells will become more important as this field progresses. To create more successful oral vaccines, research should aim to improve encapsulation techniques, develop more tools to evaluate mechanisms of mucosal immunity, and further characterize the relationship between commensals and the mucosal lymphoid tissues.

## Figures and Tables

**Figure 1 microorganisms-11-02917-f001:**
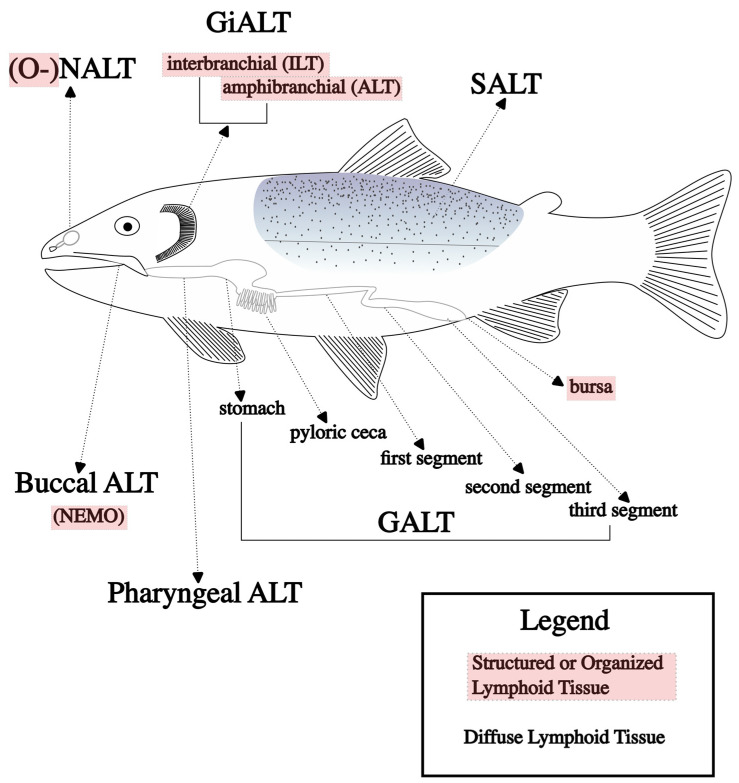
Mucosa-associated lymphoid tissues (ALTs) found in teleosts, including the specific components of the gut-associated lymphoid tissue (GALT).

**Table 1 microorganisms-11-02917-t001:** Mucosal vaccination routes and the associated lymphoid tissues stimulated by each.

Vaccination Route	Stimulated MALT		Advantages	Disadvantages	Reference
Oral	gut		Ease of delivery, which significantly reduces costs and fish stress. Allows for vaccination at multiple production stages (e.g., tanks, net pens, ponds), which increases its flexibility as a viable tool for maintaining fish health.	Less efficacious than other routes due to several factors including but not limited to the following: antigenic degradation, oral tolerance, and uneven dosing among the population.	[70]
Immersion	gut	nasal	This can easily be used to vaccinate large numbers of fish and provide a high efficacy relative to other mucosal vaccines. Stimulates multiple mucosal tissues with the use of a single delivery platform, increasing the potential for a strong immune response and increased protection against disease.	Large volumes of vaccine are required, which significantly increases the cost of vaccination. Vaccination is limited to its use in smaller fish due to the amount of vaccine required; large fish would be more difficult to handle and require substantially more volume of vaccine for a farm-scale program.	[71]
	skin	gill			
	buccal	pharyngeal			
Anal Intubation	gut		Bypasses destructive environment of the stomach, eliminating the need for encapsulation or other antigen protection. This results in a higher efficacy relative to oral delivery because the full dose is delivered to the site of antigen uptake without obstruction.	Time-consuming and induces similar stress in fish as injection vaccines, without a significant increase in protection. This is primarily a research tool to isolate and characterize the GALT response, with limited industry applicability.	[72]
Nasal	nasal		Requires less vaccine volume relative to immersion delivery and demonstrates a similar protection. This decreases the potential costs of implementing nasal vaccine programs on a farm-scale relative to immersion vaccines.	Time-consuming and labor-intensive. This is primarily a research tool to isolate and characterize the NALT response, with limited industry applicability at this time.	[66]

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
