# Peer review of "An Introduction to Relevant Immunology Principles with Respect to Oral Vaccines in Aquaculture"

_microorganisms, 2023, doi:10.3390/microorganisms11122917_

Round 1
Reviewer 1 Report
Comments and Suggestions for Authors
A succinct description of the main vaccination types used in aquaculture although there are no details on how the review was conducted which lessens the overall impact I feel. However, it is still a valuable piece of work for those interested in gaining some fundamental knowledge on the breadth of the subject.
Comments on the Quality of English Language
The review is well written and easy to read. There remain a few minor typos such as an incomplete sentence on line 312 and 'observe' rather than 'observed' on line 322.
Reviewer 2 Report
Comments and Suggestions for Authors
This systematic and complete review of the immunological principles and research progress related to oral vaccines provides important information to the reader but there is a lack of academic research progress. The authors should give some research perspectives on oral vaccines and immunity. In addition, please carefully check grammar mistakes and reference citations to meet high-quality requirements for publications.
Comments on the Quality of English LanguageThis systematic and complete review of the immunological principles and research progress related to oral vaccines provides important information to the reader but there is a lack of academic research progress. The authors should give some research perspectives on oral vaccines and immunity. In addition, please carefully check grammar mistakes and reference citations to meet high-quality requirements for publications.
Reviewer 3 Report
Comments and Suggestions for Authors
1. please review the numbers sequence,
2. the review doesn't have any hypothesis o new idea to apply in the vaccine oral administration?
3. Action mechanism from the molecular point of view?
4. Discus more about the action mechanism of oral vaccination versus immersion or injection,
5. be clear about the pros and cons of each method of vaccination,
6. why the size of fish is important before vaccination? please expand this information,
Comments on the Quality of English Language
some sentence needs clarification.
